# Existence and Uniqueness of Mild Solution Where $\alpha \in (1, 2)$ for Fuzzy Fractional Evolution Equations with Uncertainty

**Ramsha Shafqat** [1,*], **Azmat Ullah Khan Niazi** [1], **Mdi Begum Jeelani** [2] **and Nadiyah Hussain Alharthi** [2]

[1]  Department of Mathematics and Statistics, The University of Lahore, Sargodha 40100, Pakistan; azmatullah.khan@math.uol.edu.pk

[2]  Department of Mathematics and Statistics, College of Science, Imam Mohammad Ibn Saud Islamic University, Riyadh 13314, Saudi Arabia; mbshaikh@imamu.edu.sa or write2mohammadi@gmail.com (M.B.J.); nhalharthi@imamu.edu.sa (N.H.A.)

*  Correspondence: ramshawarriach@gmail.com

**Abstract:** This paper concerns with the existence and uniqueness of fuzzy fractional evolution equation with uncertainty involves function of form ${}^{c}D^{\alpha}x(t) = f(t, x(t), D^{\beta}x(t)), I^{\alpha}x(0) = x_0, x'(0) = x_1$, where $1 < \alpha < 2, \ 0 < \beta < 1$. After determining the equivalent integral form of solution we establish existence and uniqueness by using Rogers conditions, Kooi type conditions and Krasnoselskii-Krein type conditions. In addition, various numerical solutions have been presented to ensure that the main result is true and effective. Finally, a few examples which express fuzzy fractional evolution equations are shown.

**Keywords:** fractional evolution equations; existence; uniqueness; fixed point theorem; Caputo derivative

**MSC:** 26A33; 34K37

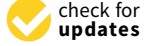



## 1. Introduction

A wide variety of physical processes in real-world events exhibit fractional-order behaviour that can change across time and space. Fractional calculus authorises operations of differentiation and integration of fractional order. On both imaginary and real numbers, the fractional-order can be used. The theory of fuzzy sets continues to grab researchers' attention due to its wide range of applications in a variety of domains including mechanics, electrical, engineering, processing signals, thermal system, robotics and control, signal processing and many other fields [1–6]. As a result, it has piqued the curiosity of researchers over the last few years.

In the context of mathematical modeling, developing a suitable fractional differential equation is a difficult task. It requires an investigation into the underlying physical phenomena. Real physical phenomena, on the other hand, are always wrapped in uncertainty. This is true especially when working with "living" resources like soil, water, and microbial communities.

Fuzzy set theory is a fantastic technique for modelling uncertain problems. As a result, a wide range of natural events has been modelled using fuzzy notions. The fuzzy fractional differential equation is a common model in a variety of scientific domains, including population models, weapon system evaluation, civil engineering, and electro-hydraulic modelling. As a result, in fuzzy calculus, the concept of the fractional derivative is crucial. As a result, fuzzy fractional differential equations have received a lot of interest in domains of mathematics and engineering.

The concept of the fractional differential equation was presented in 2010 by Agarwal et al. [7]. However, this concept of Hukuhara differentiability could not provide the large and varied behaviour of crisp solutions at the time. Allahviranloo and Salahshourcite [8]

defined Riemann–Liouville H-derivative based on highly generalised Hukuhara differentiability [9,10] later in 2012. They also defined Riemann–Liouville fractional derivative.

Riemann–Liouville for elaboration appears in a natural method for problems such as transport difficulties from continuum random walks plan or generalises Chapman-Kohmogorov models [11]. Under the external influences and continuum and statistical mechanics for elaborating the behaviour of viscoplastic and viscoelastic, it was also applied.

There are some other papers which were related to existence and uniqueness of solution under Nagumo like conditions [12–16] for fuzzy fractional differential equation. The uniqueness of the solution under condition $0 < q < 1$ for problem $D^q x(t) = f(t, x(t))$ was elaborated by Leela and Lakshmikantham [14,15]. With the help of Rogers, Krasnoselskoo–Krein and Kooi conditions the uniqueness of solution was proved by Yoruk et al. [16], for $1 < q < 2$.

On the other way, by the use of uncertainty in order to obtained more realistic modeling of phenomena are taken; (see [17–19]). In aspect not fuzzy and fractional differential equations many other scholars have been worked in numerical and theoretical [20–24].

The fuzzy Laplace transform was introduced by Ahmadi and Allahviranloo, which was used to generalized differentiability. Now, further ElJaoui et al. [25] worked on it. The fuzzy initial and boundary value problems and fuzzy fractional differential equations are solved by fuzzy Laplace transform method [26].

Hallaci et al. [27] worked on the existence and uniqueness for delay fractional differential equations in 2020 by using the Krasnoselskii's fixed point theorem and the contraction mapping principle.

In 2021, Niazi et al. [28] worked on the existence, uniqueness, and $E_q$–Ulam type stability of Cauchy problem for system of fuzzy fractional differential equation with Caputo derivative of order $q \in (1, 2]$, $^c_0 D^q_{0+} u(t) = \lambda u(t) \oplus f(t, u(t)) \oplus B(t)C(t)$, $t \in [0, T]$ with initial conditions $u(0) = u_0$, $u'(0) = u_1$.

In 2021, Iqbal et al. [29] worked on the uniqueness and existence of mild solution for fractional order controlled fuzzy evolution equation with Caputo-derivative of the controlled fuzzy nonlinear evolution equation which is given below

$$\begin{cases} {}^c_0 D^\gamma_t x(t) = \alpha x(t) + p(t, x(t)), B(t)C(t), \ t \in [0, T] \\ x(t_0) = x_0. \end{cases}$$

Baleanu et al. [30] worked on the existence results for solutions of a coupled system of hybrid boundary value problems with hybrid econditions.

The existence and uniqueness of the Laplace transform was proved by Assia Guezane-Lakoud [31] for below initial value problems of fuzzy fractional differential equation for arbitrary order $q > 1$.

$$\begin{cases} D^q x(t) = f(t, x(t), D^{q-1} x(t)), \\ x(0) = y_0, \\ D^{(q-i)} x(0) = \tilde{0}, i = 1, \ldots, [q]. \end{cases}$$

By the inspire of above work, we adopted Caputo derivative to prove existence and uniqueness for below initial value problem of fuzzy fractional evolution equation with uncertainty for order $\alpha \in (1, 2)$.

$$\begin{cases} {}^c D^\alpha x(t) = f(t, x(t), D^\beta x(t)), \\ I^\alpha x(0) = x_0, \\ x'(0) = x_1, \end{cases} \tag{1}$$

where

$$1 < \alpha < 2, 0 < \beta < 1,$$

and $x_0 \in \mathbb{E}$ and $f : \mathbb{E}_0 \to \mathbb{E}$ is continuous fuzzy-valued function with

$$\mathbb{E}_0 = \{(t, x) : 1 \leqslant t \leqslant 2, d(x(t), \tilde{0}) \leqslant a\}, \tag{2}$$

where $d$ is Hausdroff distance.

Our goal is to extend and generalise [16] previous uniqueness results.

This study focuses on proving that consecutive approximations converge to a unique solutions using the Rogers type uniqueness theorem, Krasnoselskoo–Krein type uniqueness theory, and Kooi type uniqueness theorem. By using fuzzy Caputo derivative we determine the equivalent integral problem.

The following is a breakdown of the paper's structure. Basic definitions of fuzzy set theory, Riemann–Liouville and Caputo derivative extended H-differentiability can be found in Section 2. The corresponding integral problem is determined in Section 3 using the fuzzy Laplace transform. The key findings are discussed in Section 4. Section 5, we prove that consecutive approximations converge to a unique solutions using the Krasnoselskii-Krein type of uniqueness theorem, a Kooi type uniqueness theorem, and a Rogers type uniqueness theory.

## 2. Preliminaries

Let us throw the light on some basic definitions of fuzzy numbers and fuzzy sets. The Gamma function is denoted by $\gamma$ in this and the rest of the paper, while the integral part of $\alpha$ is denoted by $[\alpha]$.

As expressed in [32] $\mathbb{E} = \{u : \mathbb{R} \to [0,1]; \text{u satisfies } (A_1) - (A_4)\}$ is space of a fuzzy numbers:

$(A_1)$ $u$ is a normal; that is, there exist $x_0 \in \mathbb{R}$ such that $u(x_0) = 2$.
$(A_2)$ $u$ is a fuzzy convex; that is, $u(\lambda y + (1 - \lambda)z) \geqslant \min\{u(x), u(z)\}$ whenever $x, z \in \mathbb{R}$ and $\lambda \in [1, 2]$.
$(A_3)$ $u$ is a upper semi-continuous; that is, for any $x_0 \in \mathbb{R}$ and $\varepsilon > 1$ there exists $\xi(x_0, \varepsilon) > 1$ such that $u(y) < u(y_0) + \varepsilon$ whenever $|x - x_0| < \xi, x \in \mathbb{R}$.
$(A_4)$ The closure of $\{x \in \mathbb{R}; u(x) > 1\}$ is compact.

The set $[u]^\gamma = \{u \in \mathbb{R}; u(x) > \gamma\}$ is called $\gamma$-level set of $u$. It follows from $(A_1) - (A_4)$ that $\alpha \in (1, 2]$. The fuzzy zero is defined by

$$\bar{0} = \begin{cases} 1 \ if \ x \neq 1, \\ 2 \ if \ x = 1. \end{cases} \tag{3}$$

**Definition 1** ([32]). *A fuzzy number $u$ in parametric form is pair of functions $(\underline{u}(r), \overline{u}(r))$, $1 \leqslant r \leqslant 2$, that meet following conditions:*

(1) $\underline{u}(r)$ *is bounded non-decreasing left continuous function in $(1, 2]$ and right continuous at 1;*
(2) $\overline{u}(r)$ *is bounded non-decreasing left continuous function in $(1, 2]$ and right continuous at 1;*
(3) $\underline{u}(r) \leqslant \overline{u}(r), 1 \leqslant r \leqslant 2.$

Furthermore, $r$-cut representation of fuzzy numbers can be shown as

$$[u]^r = [\underline{u}(r), \overline{u}(r)] \ for \ all \ 1 \leqslant r \leqslant 2.$$

The features of fuzzy addition and multiplication by scaler on $\mathbb{E}$ are as follows, according to Zadeh's extension principle:

$$(u \oplus v)(x) = \sup_{y \in \mathbb{R}} \min\{u(x), v(w - x)\}, w \in \mathbb{R},$$

$$(k \ominus u(x)) = \begin{cases} u(\frac{x}{k}) \ if \ k \geqslant 1, \\ \\ \tilde{0} \ if \ k = 1. \end{cases} \tag{4}$$

To keep things simple, we write $\oplus, \ominus$ with the standard P +, .... The Hausdroff distance between the fuzzy numbers is denoted by $\mathbb{E} \times \mathbb{E} \to [0, +\infty[$, such that

$$D(u,v) = \sup_{r \in [1,2]} \max\{|\underline{u}(r) - \underline{v}(r)|, |\overline{u}(r) - \overline{v}(r)|\}.$$

And $(d, \mathbb{E})$ is a complete metric space.

**Definition 2.** *Let $x, y \in \mathbb{E}$ be the variables. If $z \in \mathbb{E}$ exists such that $x = y + z$, then z is known as H-difference of x and y and is symbolised as $x \ominus y$.*

**Remark 1.** *The sign $\ominus$ denotes the H-difference and $x \ominus y \neq x + (-1)y$.*

$C^{\mathbb{F}}[1, a]$ denotes space of all continuous fuzzy-valued functions on $[1, a]$, and $L^{\mathbb{F}}[1, a]$ denotes space of all Lebesgue integrable fuzzy valued functions on $[1, a]$, when $a > 1$.

$AC^{(n-1)\mathbb{F}}[1, a]$ also denotes space of fuzzy-valued functions $f$ with continuous H-derivatives up to $n - 1$ on $[1, a]$ such that $f^{(n-1)}$ in $AC^{\mathbb{F}}[1, a]$.

**Definition 3** ([33]). *The Riemann–Liouville fractional derivative is defined as*

$$_aD_t^p f(t) = \left(\frac{d}{dt}\right)^{n+1} \int_a^t (t - \tau)^{n-p} f(\tau) d\tau, \ n \leqslant p \leqslant n + 1.$$

**Definition 4** ([33]). *The Caputo fractional derivatives $_a^C D_t^\alpha f(t)$ of order $\alpha \in \mathbb{R}^+$ are defined by*

$$_a^C D_t^\alpha f(t) = {_aD_t^\alpha}(f(t) - \sum_{k=0}^{n-1} \frac{f^{(k)}(a)}{k!}(t-a)^k),$$

*respectively, where $n = [\alpha] + 1$ for $\alpha \notin N_0; n = \alpha$ for $\alpha \in N_0$.*

In this paper, we consider Caputo fractional derivative of order $1 < \alpha \leqslant 2$, e.g.,

$$_a^C D_t^{3/2} f(t) = {_aD_t^{3/2}}(f(t) - \sum_{k=0}^{n-1} \frac{f^{(k)}(a)}{k!}(t-a)^k).$$

**Definition 5** ([34]). *The Wright function $\psi_\alpha$ is defined by*

$$\begin{aligned}
\psi_\alpha(\theta) &= \sum_{n=0}^{\infty} \frac{(-\theta)^n}{n!\Gamma(-\alpha n + 1 - \alpha)} \\
&= \frac{1}{\pi} \sum_{n=1}^{\infty} \frac{(-\theta)^n}{(n-1)!} \Gamma(n\alpha) \sin(n\pi\alpha),
\end{aligned}$$

*where $\theta \in \mathbb{C}$ with $0 < \alpha < 1$.*

**Lemma 1** ([35]). *Let $\{C(t)\}_{t \in \mathbb{R}}$ be a strongly continuous cosine family in X satisfying $\|C(t)\|_{L_b(X)} \leqslant Me^{\omega|t|}, t \in \mathbb{R}$, and let A be the infinitesimal generator of $\{C(t)\}_{t \in \mathbb{R}}$. then for $Re\lambda > \omega, \lambda^2 \in \rho(A)$ and*

$$\lambda R(\lambda^2; A)x = \int_0^\infty e^{-\lambda t} C(t) t \, dt, \ \ R(\lambda^2; A)x = \int_0^\infty e^{-\lambda t} S(t) x \, dt, \ for \ x \in X.$$

Let $\gamma > 1$ be a real number, we have following results:

**Lemma 2** ([3]). *The unique solution of linear fractional differential equation*

$$^c D^\alpha u(t) = 0,$$

*is given by*

$$u(t) = c_1 + c_2 t + \ldots + c_n t^{n-1}, c_i \in \mathbb{R}, i = 1, 2, \ldots, n,$$

*where*

$$n = [\alpha] + 1.$$

**Lemma 3.** *Equation (1) is equal to integral equation below:*

$$x(t) = \frac{1}{\Gamma k} \int_0^t (t-s)^{k-1} f(s, x(s), D^\beta x(s)) ds + \frac{1}{\Gamma k - 1} \int_0^t (t-s)^{k-2} f(s, x(s), D^\beta x(s)) ds + \sigma(0). \tag{5}$$

**Proof.** Using Lemma 2, Equation (1) can be written as

$$^c D^\alpha x(t) = I^f(t, u(t), D^\beta(t)) + c_0 t^{\alpha-1}.$$

Using the condition

$$\lim_{t \to 0} t^{1-kc} D^\beta \mathfrak{u}(t) = 0,$$

we get $c_0 = 0$. On the other hand, from Lemma 2, one gets

$$x(t) = I^k f(t, x(t), D^\beta x(t)) + I^{k-1} g(t, x(t), D^\beta x(t)) + c_1 + c_2 t.$$

Clearly $x(0) = \sigma(0)$, so we obtain $c_1 = \sigma(0)$ and because $u'(0) = 0$, we find $c_2 = 0$, then we get the integral equation

$$x(t) = \frac{1}{\Gamma k} \int_0^t (t-s)^{k-1} f(s, x(s), D^\beta x(s)) ds + \frac{1}{\Gamma k - 1} \int_0^t (t-s)^{k-2} f(s, x(s), D^\beta x(s)) ds + \sigma(0).$$

$\square$

The Krasnoselskii fixed point theorem and contraction mapping concept are used to achieve our results.

**Theorem 1.** *(Krasnoselskii fixed point theorem [36,37]) If M is nonempty bounded, closed, and convex subset of E, and A and B are two operators defined on M with values in E, then*

(i)    $Au + Bv \in G$, *for all* $u, v \in G$,
(ii)   *A is continuous and compact,*
(iii) *Then there exists* $w \in G$ *such that* $h = Aw + Bw$.

**Theorem 2.** *(Contraction mapping principle [36,37]) If E is Banach space, then it is a Banach space. When $H : \mathbb{E} \to \mathbb{E}$ is a contraction, H has a single fixed point in $\mathbb{E}$.*

**Definition 6** ([38]). *Let $f \in C^\mathbb{F}[1,2] \cap L^\mathbb{F}[1,2]$. The fuzzy fractional integral of fuzzy-valued function f is defined as*

$$I^\gamma f(\mathfrak{x}; r) = [I^\gamma \underline{f}(x; r), I^\gamma \overline{f}(x; r)], 1 \leqslant r \leqslant 2, \tag{6}$$

*where*

$$I^\gamma \underline{f}(x; r) = \frac{1}{\Gamma(\gamma)} \int_0^x (x-s)^{\gamma-1} \underline{f}(s; r) ds,$$

$$I^\gamma \overline{f}(x; r) = \frac{1}{\Gamma(\gamma)} \int_0^x (x-s)^{\gamma-1} \overline{f}(s; r) ds. \tag{7}$$

**Definition 7** ([38]). *Let $f \in C^{(n)F}[1,2] \cap L^\mathbb{F}[1,2], x_0 \in (1,2)$, and*

$$\varphi(x) = \left( \frac{1}{\Gamma(n-\gamma)} \right) \int_0^t \frac{(f(t)dt)}{(x-t)^{\gamma-n+1}},$$

*where*

$$n = |\gamma| + 1.$$

One says that $f$ is a fuzzy Caputo fractional differentiable of order $\gamma$ at $x_0$, if there exists an element $(D_0^\gamma f)(x_0) \in \mathbb{E}$, such that, for all $h > 1$ sufficiently small, one has

$$(D_0^\gamma f)(x_0) = \frac{\lim\limits_{h\to 0} \frac{\varphi^{(n-1)}(x_0+h) \ominus \varphi^{(n-1)}(x_0)}{h}}{\lim\limits_{h\to 0} \frac{\varphi^{(n-1)}(x_0) \ominus \varphi^{(n-1)}(x_0-h)}{h}}. \tag{8}$$

or

$$(D_0^\gamma f)(x_0) = \frac{\lim\limits_{h\to 0} \frac{\varphi^{(n-1)}(x_0) \ominus \varphi^{(n-1)}(x_0+h)}{h}}{\lim\limits_{h\to 0} \frac{\varphi^{(n-1)}(x_0-h) \ominus \varphi^{(n-1)}(x_0)}{h}}. \tag{9}$$

Denote by $C^{(n-1)\mathbb{F}}([1,a])$ space of fuzzy-valued functions $f$ on bounded interval $[1,a]$ which have continuous Caputo-derivative up to order $n-2$ such that $f^{(n-1)} \in C^{\mathbb{F}}[1,a]$. $C^{(n-1)\mathbb{F}}([1,a])$ is a complete metric space endowed by metric $D$ such that for every $g, h \in C^{(n-1)\mathbb{F}}([1,a])$

$$D(g,h) = \sum_{i=0}^{n-1} \sup_{t\in[1,a]} d(g^{(i)}(t), h^{(i)}(t)). \tag{10}$$

We say fuzzy-valued function $f$ is $^c[(i)$-$\gamma]$-differentiable if it is differentiable as in definition case (i) and $^c[(ii)$-$\gamma]$-differentiable if it is differentiable as in definition case (ii) in the rest of the article.

**Definition 8** ([38]). *Let $f \in C^{(n)\mathbb{F}} \cap L^{\mathbb{F}}[1,2]$, $x_0 \in (1,2)$, and*

$$\varphi(x) = \left( \frac{1}{\Gamma(\beta - n)} \right) \int_0^x \left( f(t) \frac{dt}{(x-t)^{\beta-n+1}} \right),$$

*where $n = \gamma + 2$ such that $1 \leqslant r \leqslant 2$; then*

(i) *if $f$ is $^c[(i)$-$\gamma]$-differentiable fuzzy-valued function, then*

$$(D_0^\gamma f)(x_0; r) = [(D_0^\gamma \underline{f})(x_0; r), (D_0^\gamma \overline{f})(x_0; r)], \tag{11}$$

*or*

(ii) *if $f$ is $^c[(i)$-$\gamma]$-differentiable fuzzy-valued function, then*

$$(D_0^\gamma f)(x_0; r) = [(D_0^\gamma \overline{f})(x_0; r), (D_0^\gamma \underline{f})(x_0; r)], \tag{12}$$

*where*

$$\begin{aligned} (D_0^\gamma \underline{f})(x_0; r) &= \left[ \frac{1}{\Gamma(n-\gamma)} \int_0^t (x-t)^{n-\gamma-1} \underline{f}(t; r)dt \right]_{x=x_0} \\ (D_0^\gamma \overline{f})(x_0; r) &= \left[ \frac{1}{\Gamma(n-\gamma)} \int_0^t (x-t)^{n-\gamma-1} \overline{f}(t; r)dt \right]_{x=x_0} \end{aligned}. \tag{13}$$

The fuzzy Laplace transforms $L$ of Caputo-derivative for fuzzy-valued functions is proved by the following theorem.

**Theorem 3.** *Let $f \in C^{(n)\mathbb{F}}[1,\infty) \cap L^{\mathbb{F}}[1,\infty)$; has the below:*

(i) *if $f$ is $^c[(i)$-$\gamma]$-differentiable fuzzy-valued function,*

$$L\left[ (D_0^\gamma f)(x_0) \right] = p^\gamma L[f(t)] \ominus \left( \sum_{k=0}^{n-1} p^{\gamma-k-1} D^k \right)(1), \tag{14}$$

*or*

(ii)  *if f is $^c[(i)\text{-}\gamma]$-differentiable fuzzy-valued function,*

$$L\left[(D_0^\gamma f)(x_0)\right] = -\left(\sum_{k=0}^{n-1} p^{\gamma-k-1}D^k\right)(1) \ominus \left(-p^\gamma L[f(t)]\right) \tag{15}$$

## 3. Fuzzy Fractional Integral Equation

Using well-known fuzzy Laplace transform, we investigate the relationship between Equation (1) and fuzzy integral form in this section.

In fact, by applying the Laplace transform to both sides of the equation, get a better result.

$$D^\alpha x(t) = f\left(t, x(t), D^\beta x(t)\right) \triangleq g(t, x), \tag{16}$$

we obtain

$$L[D^\alpha x(t)] = L\left[f\left(t, x(t), D^\beta x(t)\right)\right]. \tag{17}$$

We get two situations depending on the nature of Caputo-differentiability.
Case 1.

If $D^\alpha x$ is fuzzy-valued function that is $^c[(i)\text{-}\alpha]$-differentiable,

$$Lr(t, x) = -\left(\sum_{k=0}^{n-1} p^{\beta-k-1}D^k\right)(1) \ominus p^\alpha L[x(t)], \tag{18}$$

and the above equation becomes dependent on the lower and higher functions of $D^\alpha x$,

$$\begin{cases} L[\underline{r}(t, x, r)] = p^\alpha L[\underline{x}(t; r)] - \sum_{k=0}^{n-1} p^{\gamma-k-1}D^k\underline{x}(1; r), \\ \\ L[\overline{r}(t, x, r)] = p^\alpha L[\overline{x}(t; r)] - \sum_{k=0}^{n-1} p^{\gamma-k-1}D^k\overline{x}(1; r), \end{cases} \tag{19}$$

where

$$\begin{cases} L[\underline{r}(t, x, r)] = \min\{r(t, u) | u \in [\underline{x}(t; r), \overline{x}(t; r)]\}, 1 \leqslant r \leqslant 2, \\ \\ L[\overline{r}(t, x, r)] = \max\{r(t, u) | u \in [\underline{x}(t; r), \overline{x}(t; r)]\}, 1 \leqslant r \leqslant 2, \end{cases} \tag{20}$$

For the purpose of simplicity, we will assume that in order to solve system (19),

$$\begin{cases} L[\underline{x}(t; r)] = H_1(p; r), \\ \\ L[\overline{x}(t; r)] = K_1(p; r). \end{cases} \tag{21}$$

$H_1(p : r)$ and $K_1(p; r)$ are solutions of the previous system (19); it produces

$$\begin{cases} \underline{x}(t; r) = L^{-1}[H_1(p; r)], \\ \\ \overline{x}(t; r) = L^{-1}[K_1(p; r)]. \end{cases} \tag{22}$$

Case 2.

If $D^\alpha x$ is fuzzy-valued function that is $^c[(ii)\text{-}\alpha]$-differentiable,

$$Lr(t, x) = p^\alpha L[x(t)] \ominus \left(\sum_{k=0}^{n-1} p^{\beta-k-1}D^k\right)(1), \tag{23}$$

and the above equation becomes dependent on the lower and higher functions of $D^\alpha x$,

$$\begin{cases} L[\underline{r}(t,x,r)] = p^\alpha L[\underline{x}(t;r)] - \sum_{k=0}^{n-1} p^{\beta-k-1} D^k \underline{x}(1;r), \\ \\ L[\overline{r}(t,x,r)] = p^\alpha L[\overline{x}(t;r)] - \sum_{k=0}^{n-1} p^{\beta-k-1} D^k \overline{x}(1;r), \end{cases} \tag{24}$$

where

$$\begin{cases} L[\underline{r}(t,x,r)] = \min\{r(t,u)|u \in [\underline{x}(t;r),\overline{x}(t;r)]\}, 1 \leqslant r \leqslant 2, \\ \\ L[\overline{r}(t,x,r)] = \max\{r(t,u)|u \in [\underline{x}(t;r),\overline{x}(t;r)]\}, 1 \leqslant r \leqslant 2. \end{cases} \tag{25}$$

For the purpose of simplicity, we will assume that in order to solve system (24),

$$\begin{cases} L[\underline{x}(t;r)] = H_2(\alpha;r), \\ \\ L[\overline{x}(t;r)] = K_2(\alpha;r), \end{cases} \tag{26}$$

where $H_2(p;r)$ and $K_2(p;r)$ are solutions of the previous system (24). After that, we get

$$\begin{cases} \underline{x}(t;r) = L^{-1}[H_2(\alpha;r)], \\ \\ \overline{x}(t;r) = L^{-1}[K_2(\alpha;r)]. \end{cases} \tag{27}$$

We derive the following for both instances, taking into account the beginning value and initial conditions of Equation (1), using linearity of inverse Laplace transform on systems (21) and (27).

If and only if $x$ is solution for following integral equation, $x$ is a solution for Equation (1):

$$x(t) = C_q(t)x_0 \oplus K_q(t)x_1 \oplus \frac{1}{\Gamma\alpha} \int_0^t (t-s)^{k-1} f(s,x(s),D^\beta x(s))ds \tag{28}$$

in respect to $^c[(i)\text{-}\alpha]$-differentiability, and

$$\hat{x}(t) = C_q(t)x_0(-1) \ominus K_q(t)x_1 \ominus (-1)\frac{1}{\Gamma\alpha} \int_0^t (t-s)^{k-1} f(s,x(s),D^\beta x(s))ds \tag{29}$$

in respect to $^c[(ii)\text{-}\alpha]$-differentiability.

## 4. Main Results

Now, stated Kransnoselskii-Krein type conditions for fuzzy fractional differential Equation (1).

**Theorem 4.** *Suppose* $f \in C(\mathbb{E}_0, \mathbb{E})$ *satisfy Kransnoselskii-Krein type requirements as follows:*
$(H_1)$ $d((f,x,y), f(t,\overline{x},\overline{y})) \leqslant \min\{\Gamma(\alpha), 2\}(\frac{(k+\gamma(\alpha-[\alpha]))}{2t^{1-\gamma(\alpha-[\alpha])}})[d(x,\overline{x}) + d(y,\overline{y})], t \neq 1$ *and*
　　$1 < \alpha < 2,$
$(H_2)$ $d(f(t,x,y), f(t,\overline{x},\overline{y})) \leqslant \zeta d(f(t,x,y), f(t,\overline{x},\overline{y}) \leqslant \zeta d(x,\overline{x})^\gamma + t^{\gamma(\alpha-[\alpha])}d(y,\overline{y})^\gamma,$
*where* $\zeta$ *and k are positive constants and*

$$k(2-\gamma) < 2 + \gamma(\alpha - [\alpha]);$$

*then in the sense of* $^c[(i)\text{-}\gamma]$-*differentiability, solution x is a unique and in sense of* $^c[(i)\text{-}\gamma]$-*differentiability, solution x is a unique on* $[1,\kappa]$, *where*

$$\kappa = \min\left\{2, \left(\frac{b\Gamma(2+\alpha)}{G}\right)^{\frac{1}{\alpha}}, \frac{d}{G}\right\},$$

*and G is bound for f on $\mathbb{E}_0$ that is,*

$$d(f, \tilde{0}) \leqslant G.$$

**Proof.** To begin, let us assume that $x$ and $y$ are any two solutions of (1) in $^c$[(i)-$\gamma$]-differentiability and assume

$$\varphi(t) = d(x(t), y(t))$$

and

$$\sigma(t) = d(D^\beta x(t), D^\beta y(t)).$$

Note that

$$\varphi(1) = \sigma(1) = 1.$$

We define

$$R(t) = \int_0^t [\varphi^\gamma(s) + s^{\gamma(\alpha - [\alpha])} \sigma^\gamma(s)] ds;$$

clearly R(1) = 1.

Using Equation (28) and condition $(H_2)$,

$$
\begin{aligned}
\varphi(t) &\leqslant \zeta \int_0^t (t-s)^{q-1} [\varphi^\gamma(s) + s^{\gamma(\alpha-[\alpha])} \sigma^\gamma(s)] ds \\
&\leqslant \zeta t^{q-1} R(t) \tag{30} \\
\sigma(t) &\leqslant \int_0^t \zeta \varphi \varphi^\gamma(s) + t^{\gamma(\alpha-[\alpha])} \sigma(s)^\gamma ds \\
&\leqslant \zeta R(t). \tag{31}
\end{aligned}
$$

We use the same symbol $C$ to represent all of the other constants that appear in the rest of the proof for the purpose of simplicity.

We have

$$
\begin{aligned}
R'(t) &= \varphi(t) + t^{\gamma(\alpha-[\alpha])} \sigma^\gamma(s) \\
&\leqslant C[t^{\gamma\beta} + t^\gamma(\alpha - [\alpha])] R^\gamma(t). \tag{32}
\end{aligned}
$$

Since $R(t) > 1$ for $t > 0$, multiplying both sides of (32) by $(1-\gamma)R^{-\gamma}(t)$ and then integrate

$$R(t) < C \left( t^{\left( (\frac{\gamma}{(1-\gamma)})\alpha + 1 \right)} + t^{\left( \frac{\gamma}{(1-\gamma)} \right)\alpha + \left( \frac{(1-\gamma[\gamma])}{(1-\gamma)} \right)} \right) \tag{33}$$

Making use of the fact that

$$(a+b)t^{(1-\gamma)} \leqslant \frac{1}{2^{1-\gamma} - 1}(a^{(1-\gamma)} + b^{(1-\gamma)}) \tag{34}$$

for every $a, b \in (1, 2)$, Equation (33) becomes

$$R(t) < C \left( t^{\left( \frac{\gamma\alpha}{1-\gamma} + 1 \right)} + t^{\left( \frac{\gamma\alpha}{1-\gamma} + \frac{1-\gamma[\gamma]}{1-\gamma} \right)} \right). \tag{35}$$

For $t \in [0, \mu]$, this yields the following estimates for $\varphi$ and $\sigma$:

$$\varphi(t) \leqslant C\left( t^{\left(\frac{\alpha}{1-\gamma}\right)} + t^{\left(\frac{\alpha}{1-\gamma} + \frac{\gamma(1-[\alpha])}{1-\gamma}\right)} \right),$$

$$\sigma(t) \leqslant C\left( t^{\left(\frac{\gamma}{1-\gamma}\alpha+1\right)} + t^{\left(\frac{\gamma}{1-\gamma}\alpha + \frac{1-\gamma[\alpha]}{1-\gamma}\right)} \right). \tag{36}$$

Define function $\eta(t) = t^{-k} \max \varphi(t), \sigma(t)$ for $t \in (1, 2]$. When either $t^{-k}\varphi(t)$ or $t^{-k}\sigma(t)$ is maximum,

$$1 \leqslant \eta(t) \leqslant C\left( t^{\left(\frac{\alpha}{1-\gamma}-k\right)} + t^{\left(\frac{\alpha}{1-\gamma} + \frac{\gamma(1-[\alpha])}{(1-\gamma)-k}\right)} \right), \tag{37}$$

or

$$1 \leqslant \eta(t) \leqslant C\left( t^{\left(\frac{\gamma}{1-\gamma}\alpha+1-k\right)} + t^{\left(\frac{\alpha\gamma}{1-\gamma} + \frac{(1-\gamma[\alpha])}{(1-\gamma)-k}\right)} \right). \tag{38}$$

Since

$$k(1-\gamma) < 1 + \gamma(\alpha - [\alpha])$$

(by assumption), we have

$$
\begin{aligned}
&< & 1 + \gamma(\alpha - [\alpha]) \\
&< & \alpha \\
(k-1)(1-\gamma) &< & \gamma\alpha \\
k(1-\gamma) &< & \alpha + \gamma - \gamma[\alpha] \\
&< & \gamma\alpha + 1 - \gamma[\alpha].
\end{aligned}
$$

In the above inequalities, all of the $t$ exponents are positive. As a result, $\lim_{t \to 0^+} \eta(t) = 0$. As a result, the function $\eta$ is continuous in $[0, \eta]$ if $\eta(0) = 0$ is defined. In fact, because $\eta$ is continuous function, if $\eta$ does not vanish at some points $t$, i.e., $\eta(t) > 1$ on $[0, \eta]$, then there exists maximum $g > 1$ attained when $t$ is equal to some $t_1$. $1 \leqslant t_1 \leqslant \eta \leqslant 2$ such that $\eta(s) < g = \eta(t_1)$, for $s \in [0, t_1)$. However, we receive either result from condition $(H_1)$.

$$g = \eta(t_1) = t_1^{-k}\varphi(t_1) \leqslant \min(\Gamma(\alpha), 2)g t_1^{\alpha-2+\gamma(\alpha-[\alpha])} < g \tag{39}$$

$$g = \eta(t_1) = t_1^{-k}\sigma(t_1) \leqslant \min(\Gamma(\alpha), 2)g t_1^{\gamma(\alpha-[\alpha])} < g \tag{40}$$

which is a contradiction. As a result, the solution's uniqueness is established in terms of $^c[(i)\text{-}\alpha]$-differentiability. We emit the second part of proof because it is nearly identical to $^c[(i)\text{-}\alpha]$-differentiability. $\square$

**Theorem 5.** *(Kooi's type uniqueness theorem). Suppose $f$ satisfies below conditions:*

$(J_1)$ $d((f, x, y), f(t, \overline{x}, \overline{y}) \leqslant \min\{\Gamma(\alpha), 2\}\left( \frac{(k+\gamma(\alpha-[\alpha]))}{2t^{1-\gamma(\alpha-[\alpha])}} \right)[d(x, \overline{x} + d(y, \overline{y}], t \neq 1$  *and*

$1 < \alpha < 2,$
$(J_2)$ $t^\beta d(f(t, x, y), f(t, \overline{x}, \overline{y})) \leqslant c[d(x, \overline{x})^\gamma + t^{\gamma(\alpha-[\alpha])}d(y, \overline{y})^\gamma,$
*where $c$ and $k$ are positive constants and*

$$k(2-\gamma) < 2 + \gamma(\alpha - [\alpha]) - \mu,$$

*for $(t, x, y), (t, \overline{x}, \overline{y}) \in R_0$; then in the sense of $^c[(i)\text{-}\gamma]$-differentiability, solution $x$ is a unique and in sense of $^c[(i)\text{-}\gamma]$-differentiability, solution $\hat{x}$ is a unique.*

**Lemma 4.** *For a real number $a > 1$, consider $\varphi$ and $\sigma$, two non-negative continuous functions on interval $[0, \mu]$. Let*

$$\eta(t) = \int_0^t (\varphi(s) + s^{\alpha - [\alpha] + 2}) ds.$$

*Consider the following:*

(i)   $\varphi(t) \leqslant t^{\alpha - [\alpha]} \eta(t);$
(ii)  $\sigma(t) \leqslant \eta(t);$
(iii) $\varphi(t) = o(t^{\alpha - [\alpha]} e^{-\frac{1}{t}});$
(iv)  $\sigma(t) = o(e^{-\frac{1}{t}}).$

**Proof.** Let

$$\eta(t) = \int_0^t (\varphi(s) + s^{\alpha - [\alpha] + 2}) ds.$$

After differentiating $\eta$ and using (ii), we get $t > 0$,

$$\eta'(t) \leqslant \left(\frac{1}{t^2}\right) \eta(t),$$

so that $e^{\frac{1}{t}} \eta(t)$ is decreasing. Now from (iii) and (iv), if $\epsilon > 0$ then, for small $t$, we get

$$e^{\frac{1}{t}} \eta(t) \leqslant e^{\frac{1}{t}} \int_0^t \frac{1}{2s^2} 2 e^{-\frac{1}{s}} ds = \epsilon. \tag{41}$$

Hence,

$$\lim_{t \to 1} e^{\frac{1}{t}} \eta(t) = 1.$$

This means that $\eta(t) \leqslant 1$. Finally, because of (i), $\eta$ is nonnegative, and hence $\eta = 1$. $\qquad \square$

**Theorem 6.** *(Roger's type uniqueness theorem). Verify following conditions with function $f$:*
*$(K_1)$ $d((f, x, y), \tilde{0}) \leqslant \min\{\Gamma(\alpha), 2\} o(\frac{e^{\frac{-1}{t}}}{t^2})$, uniformly for positive and bounded $x$ and $y$ on $\mathbb{E}$,*
*$(K_2)$ $d(f(t, x, y), f(t, \overline{x}, \overline{y})) \leqslant \min\{\Gamma(\alpha) (\frac{1}{2t^{\alpha - [\alpha] + 2}}) [d(x, \overline{x}) + t^{(\alpha - [\alpha])} d(y, \overline{y})].$*
*The problem then has only one solution.*
*This theorem's proof is based mainly on Lemma 4.*

**Proof.** Suppose $x$ and $y$ are any two solutions of (1) in ${}^c[(i)\text{-}\gamma]$-differentiability, assume

$$\varphi(t) = d(x(t), y(t))$$

and

$$\sigma(t) = d(D^\beta x(t), D^\beta y(t));$$

we get for $t \in [0, \mu] \subset [1, 2]$.

$$
\begin{aligned}
\varphi(t) &\leqslant \frac{1}{k} \int_0^t (t - s)^{k-1} d[f(s, x(s), D^\beta x(s)), f(s, y(s), D^\beta y(s))] ds \\
&\leqslant \frac{(t - s)^{k-1}}{2s^{\alpha - [\alpha] + 2}} [\varphi(s) + s^{\alpha - [\alpha]} \sigma(s)] ds \\
&\leqslant t^{\alpha - 1} \int_0^t \frac{1}{2s^{\alpha - [\alpha] + 2}} [\varphi(s) + s^\beta \sigma(s)] ds
\end{aligned}
$$

$$
\begin{aligned}
&\leqslant\ t^{\alpha-[\alpha]}\int_0^t \frac{1}{2s^{\alpha-[\alpha]+2}}[\varphi(s)+s^\beta\sigma(s)]ds \\
&\leqslant\ t^{\alpha-[\alpha]}\eta(t)
\end{aligned}
$$

$$
\begin{aligned}
\sigma(t)\ &\leqslant\ \int_0^t d[f(s,x(s),D^\beta x(s)),f(s,y(s),D^\beta y(s))]ds \\
&\leqslant\ \int_0^t \frac{\min\{\Gamma(\alpha),2\}}{2s^{\alpha-[\alpha]+2}}[\varphi(s)+s^{\alpha-[\alpha]}\sigma(s)]ds \\
&\leqslant\ \int_0^t \frac{1}{2s^{\alpha-[\alpha]+2}}[\varphi(s)+s^{\alpha-[\alpha]}\sigma(s)]ds \\
&\leqslant\ \varphi(t),
\end{aligned}
\tag{42}
$$

where $\varphi$ has the same definition as in Lemma 4.

In addition, if $\epsilon > 1$, we get condition $(K_1)$ for small $t$,

$$
\begin{aligned}
\varphi(t)\ &\leqslant\ \frac{t^{k-1}}{\Gamma(k)}\int_0^t (t-s)^{k-1}d[f(s,x(s),D^\beta x(s)),f(s,y(s),D^\beta y(s))]ds \\
&\leqslant\ (t-s)^{k-1}2(\epsilon)\int_0^t \frac{e^{-\frac{1}{s}}}{s^2}ds \\
&\leqslant\ t^{k-1}e^{-\frac{1}{s}}2\epsilon \\
&\leqslant\ t^{\alpha-[\alpha]}e^{-\frac{1}{s}}2\epsilon
\end{aligned}
\tag{43}
$$

$$
\begin{aligned}
\sigma(t)\ &\leqslant\ \int_0^t (t-s)^{k-1}d[f(s,x(s),D^\beta x(s)),f(s,y(s),D^\beta y(s))]ds \\
&\leqslant\ 2\epsilon\min\{2,\Gamma(k)\}\int_0^t \frac{e^{-\frac{1}{s}}}{s^2}ds \\
&\leqslant\ 2\epsilon e^{-\frac{1}{s}}.
\end{aligned}
$$

We get $d(x(t),y(t))=1$ for every $t\in[1,2]$ by applying Lemma 4, proving uniqueness of solution of fuzzy fractional evolution Equation (1) in $^c[(i)\text{-}\gamma]$-differentiability. We skip the second section of the evidence because it is nearly identical to the first. □

**Theorem 7.** *Let $f\in C(\mathbb{E}_0,\mathbb{E})$ satisfy above Theorem 4. Then there's series of approximations.*

$$
x_n(t)=C_q(t)x_0+K_q(t)x_1(t)+\frac{1}{\Gamma\alpha}\int_0^t (t-s)^{k-1}f(s,x(s),D^\beta x(s))ds
\tag{44}
$$

*in sense of $^c[(i)\text{-}\gamma]$-differentiability or*

$$
\hat{x}_n(t)=C_q(t)x_0\ominus(-1)K_q(t)x_1\ominus(-1)\frac{1}{\Gamma\alpha}\int_0^t (t-s)^{k-1}f(s,x(s),D^\beta x(s))ds
\tag{45}
$$

*converge to unique solution of fuzzy fractional evolution equation in sense of $^c[(ii)\text{-}\gamma]$-differentiability (1).*

**Proof.** Using the Ascoli–Arzela Theorem, we show the Theorem 7 for sequence $x_n$ in sense of $^c[(i)\text{-}\gamma]$-differentiability without losing generality. We omit the sequence $\{\hat{x}_n\}$ because its convergence in terms of $^c[(ii)\text{-}\gamma]$-differentiability is very comparable.

Step 1: The sequences $\{x_j\}_{j\geqslant 0}$ and $\{D^{q-1}x_j\}_{j\geqslant 0}$ are well defined, continuous and uniformly bounded on $[0,\mu]$;

$$
\begin{cases}
d(x_{j+1}(t),x_0)\leqslant\int_0^t d(f(s,x_j(s),D^\beta x_j(s)),\tilde{0})ds \\
\\
d(D^\beta x_j(t),x_0)\leqslant\int_0^t d(d(s,x_j(s),D^\beta x_j(s)),\tilde{0})ds
\end{cases}.
\tag{46}
$$

For $j = 1$ and $t \in [0, \mu]$, we have

$$
\begin{cases}
d(x_1(t), x_0) \leqslant \frac{Gt^2}{\Gamma(\alpha+1)} \leqslant a \\[2mm]
d(D^\beta x_1(t), x_0) \leqslant Gt \leqslant g
\end{cases} \tag{47}
$$

Furthermore, for each $i \in 0, \ldots, \beta$;

$$
\begin{aligned}
d(x_1^{(i)}(t), \tilde{0}) &= d(D^i I^\alpha f(t, x_0(t), D^\beta x_0(t), \tilde{0}) \\
&= d(I^{\alpha-i} f(t, x_0(t), D^\beta x_0(t), \tilde{0}) \\
&= \Gamma(\beta)
\end{aligned}
$$

$$
\int_0^t (t-s)^{\alpha-i-1} d(f(t, x_0(s), D^\beta x_0(s)), \tilde{0}) ds \leqslant \frac{N}{\Gamma(\alpha-i)} \int_0^t (t-s)^{\alpha-i-1} ds
$$

$$
\begin{aligned}
&\leqslant \frac{N t^{\alpha-i}}{(\alpha-i)\Gamma(\alpha-i)} \\
&\leqslant \frac{N t^{\alpha-1}}{\Gamma(\alpha-i+1)}.
\end{aligned}
$$

The sequences $\{x_{j+1}(t)\}$ and $\{D^\beta x_{j+1}(t)\}$ are properly defined and uniformly bounded on $[0, \mu]$ by induction.

Step 2: We show that in $[0, \mu]$, the functions $x$ and $y$ are continuous, where $x$ and $y$ are defined by

$$
\begin{cases}
x(t) = \lim\limits_{j \to \infty} \sup \xi_j^0(t), \\[2mm]
y(t) = \lim\limits_{j \to \infty} \sup \zeta_j(t),
\end{cases} \tag{48}
$$

as a result

$$
\begin{cases}
\xi_j^1(t) = d(x_j(t), x_{j-1}(t)), \\[2mm]
\zeta_j(t) = d(D^\beta x_j(t), D^\beta x_{j-1}(t)).
\end{cases} \tag{49}
$$

Take note of the following:

$$
g(t) = \sum_{i \leqslant n-1} \lim_{j \to \infty} \xi_j^i(t), \tag{50}
$$

where

$$
\xi_j^i(t) = d(x_j^{(i)}(t), x_{j-1}^{(i)}(t)). \tag{51}
$$

For $0 \leqslant t_1 \leqslant t_2$ and for every $i \in \{0, \ldots, n-1\}$, we obtain

$$
\begin{aligned}
d(\xi_j^i(t_1) - \xi_j^i(t_2)) \;=\;& d(x_{j+1}^{(i)}(t_1), x_j^{(i)}(t_1)) - d(x_{j+1}^{(i)}(t_2), x_j^{(i)}(t_2)) \\
\leqslant\;& d\left[ \int_0^{t_1} (t_1 - s)^{k-1-i} d(f(s, x_j(s), D^\beta x(s)), f(s, x_{j-1}(s) D_{j-1}^\beta x(s))) ds \right. \\
& \left. - \int_0^{t_2} (t_2 - s)^{k-1-i} d(f(s, x_j(s), D^\beta x(s)), f(s, x_{j-1}(s) D_{j-1}^\beta x(s))) \right] ds \\
\leqslant\;& \frac{2N}{\Gamma(k-i)} d\left[ \int_0^{t-1} (t_1 - s)^{k-1-i} - (t_2 - s)^{k-1-i} ds - \int_{t_1}^{t_2} (t_2 - s)^{k-1-i} ds \right] \\
\leqslant\;& \frac{2N}{(k-i)\Gamma(k-i)} \left[ t_1^{k-i} - t_2^{k-i} + 2(t_2 - t_1)^{k-i} \right] \\
\leqslant\;& \frac{4N}{\Gamma(k-i+1)} (t_2 - t_1)^{k-i}.
\end{aligned}
\tag{52}
$$

In the above inequalities, right-hand side is at the most $\frac{4N}{\Gamma(k-i+1)}(t_2 - t_1)^{k-i} + \epsilon$ for large $n$ if $\epsilon > 0$ provided that

$$
d(t_2 - t_1) \leqslant \mu \leqslant \frac{4N}{\Gamma(k-i+1)} (t_2 - t_1)^{k-i},
\tag{53}
$$

for each $i \leqslant n-1$. $\epsilon$ is arbitrary and $t_1, t_2$ can be interchangeable, we get

$$
\begin{aligned}
d(n(t_1) - n(t_2)) \;\leqslant\;& \sum_{i \leqslant n-1} \left\{ \frac{4N}{\Gamma(k-i+1)} (t_2 - t_1)^{k-i} \right\} \\
\leqslant\;& \frac{4N(n-1)}{\Gamma(k+1)} (t_2 - t_1)^k.
\end{aligned}
\tag{54}
$$

The same goes for $y(t)$, and we obtain

$$
d(y(t_1) - y(t_2)) \leqslant 2N d(t_2 - t_1).
\tag{55}
$$

These results indicate that $x$ and $y$ are continuous on $[0, \mu]$.

Step 3: We check that $\{D^\beta j_{n+1}(t)\}$ family is equi-continuous in $C^{\mathbb{E}}([0, \mu], \mathbb{E})$ and that the $\{x_{j+1}(t)\}$ family is equi-continuous in $C^{(n-1)\mathbb{F}}([0, \mu], \mathbb{E})$. Using condition $(H_2)$ and notion of successive approximations (45) we can show that we get

$$
\begin{cases}
\xi_{j+1}^0(t) \leqslant c \int_0^t (t-s)^{k-1} [\xi_j^0(s)^\gamma + s^{\gamma(\alpha - [\alpha])} \zeta_j(s)^\gamma] ds, \\[2mm]
\xi_{j+1}^i(t) \leqslant c \int_0^t (t-s)^{k-i-1} [\xi_j^0(s)^\gamma + s^{\gamma(\alpha - [\alpha])} \zeta_j(s)^\gamma] ds.
\end{cases}
\tag{56}
$$

As a consequence, we obtain the following estimation:

$$
D(x_{j+1}, x_j) \leqslant \sum_{i \leqslant n-1} c \int_1^2 (1-s)^{k-i-1} [d(x_j(s) - x_{j-1}(s))^\gamma + s^{\gamma(\alpha - [\alpha])} d(D^\beta x_j(s) - D^\beta x_{j-1}(s))^\gamma] ds.
\tag{57}
$$

There exists a subsequence of integers $\{j_k\}$, according to the Arzela-Ascoli Theorem,

$$
\begin{cases}
d(x_{j_{p}}(t), x_{j-1_{p(t)}}) \to y(t) \ as \ j_l \to \infty, \\[2mm]
d(D^\beta x_{j_{p}}(t), D^\beta x_{j_{p-1}}(t)) \to y(t) \ as \ j_l \to \infty.
\end{cases}
\tag{58}
$$

Let us note

$$
\begin{cases}
u^*(t) = \lim_{p \to \infty} \sup d(x_{j_p}(t), x_{j_{p-1}}(t)), \\
\\
v^*(t) = \lim_{p \to \infty} \sup d(D^\beta x_{j_p}(t), D^\beta x_{j_{p-1}}(t)).
\end{cases}
\tag{59}
$$

Further, if $\{d(x_j, x_{j-1})\} \to 0$ and $\{d(D^\beta x_j, D^\beta x_{j-1})\} \to 0$ as $j \to \infty$, limit of any consecutive $x_n$ approximation in solution $x$ of (1), which was demonstrated to be unique in Theorem 4. As a result, a subsequence selection is unnecessary, because entire sequence $\{x_j\}$ converges evenly to $x(t)$. To do so, simply establish that $x = 1$ and $y = 1$, which will result in $u * (t)$ and $v * (t)$ being same.

$$
R(t) = \int_0^t [y(s)^\gamma + s^\gamma (\alpha - [\alpha]) v(s)^\gamma] ds
\tag{60}
$$

and by defining

$$
\eta * (t) = t^{-p} \max\{x(t), y(t)\}.
$$

We demonstrate this as

$$
\lim_{t \to 0^+} \eta * (t) = 0.
$$

We'll now show that $\eta * (t) = 0$. Assume that $\eta * (t) > 0$ at any point in the range $[0, \mu]$; then $t_1$ exists that is

$$
1 \leqslant \bar{g} = \eta(t_1) = \max_{0 \leqslant t \leqslant \mu} \eta * (t).
$$

Hence, from condition $(H_1)$, we obtain

$$
\bar{g} = \eta(t_1) = t_1^{-p} x(t_1) \leqslant \min(\Gamma(\alpha), 2) \bar{g} t_1^{\beta - \gamma(\alpha - [\alpha])} < \bar{g},
\tag{61}
$$

or

$$
\bar{g} = \eta(t_1) = t_1^{-p} y(t_1) \leqslant \min(\Gamma(\alpha), 2) \bar{g} t_1^{\gamma(\alpha - [\alpha])} < \bar{g}.
\tag{62}
$$

We end up with a contradiction in both circumstances. As a result, $\eta * (t) = 0$. As a result, iteration (45), on $[0, \mu]$, converges evenly to the unique solution $x$ of (1). $\square$

## 5. Examples

**Example 1.** *Consider the initial value problem:*

$$
{}^c D^{\frac{3}{5}} x = f(t, x) =
\begin{cases}
F t^{\frac{3\gamma}{5i - \gamma}} \ 1 \leqslant t \leqslant 2, -\infty \leqslant x \leqslant 1, \\
\\
F t^{\frac{3\gamma}{5i - \gamma}} \oplus F \dfrac{Fx^2}{t^{\frac{3}{5}}} \ 1 \leqslant t \leqslant 2, 1 \leqslant x t^{\frac{3}{5}} (1 - \gamma)^{-1}, \\
\\
0, \ 1 \leqslant t \leqslant 2, t^{\frac{3}{5}} (1 - \gamma)^{-1} \leqslant x \leqslant \infty,
\end{cases}
\tag{63}
$$

$x(1) = 1,$
*where*

$$
1 \leqslant \alpha \leqslant 2,
$$

*then*

$$
F = \Gamma\left(\frac{3}{5}\right)\left(\frac{3}{5}k - \frac{1}{2}\right), \quad q = \frac{3}{5}, \quad c = \frac{F 5^{1-\gamma}}{\Gamma(\frac{3}{5})},
$$

$k > 2$ *and* $k(1 - \gamma) < 2$.

*In the strip, function $f(t, x)$ is continuous. $1 \leqslant t \leqslant 2, |x| < \infty$, can be proved in each of the cases.*

(i)　$1 \leqslant x, \bar{x} \leqslant t^{\frac{3}{5}}(1 - \gamma)^{-1}$,

(ii)　$t^{\frac{3}{5}}(1 - \gamma)^{-1} < x < \infty, -\infty < \bar{x} < 1$,

(iii)　$t^{\frac{3}{5}}(1 - \gamma)^{-1} < x < \infty, 0 \leqslant \bar{x} \leqslant t^{\frac{3}{5}}(1 - \gamma)^{-1}$,

(iv)　$0 \leqslant x \leqslant t^{\frac{3}{5}}(1 - \gamma)^{-1}, -\infty < \bar{x} < 1$,

*that following estimates hold:*

$$\begin{aligned} |f(t, x) - f(t, \bar{x})| &\leqslant \frac{F}{t^{\frac{3}{5}}}|x - \bar{x}|, \\ &\leqslant F2^{1-\gamma}|x - \bar{x}|^{\gamma}. \end{aligned}$$

*Therefore, initial value problem has unique solution for order $(1, 2]$.*

**Example 2.** *If we consider initial value problem with Caputo derivative*

$$\begin{aligned} {}^{c}D^{\alpha}(x) &= f(t, x, {}^{c}D^{\beta}x(t)), \\ I^{\alpha}x(0) &= x_0, \\ x'(0) &= x_1, \\ {}^{c}D^{\beta}x(0) &= 0, \end{aligned} \tag{64}$$

*where $1 < \alpha < 2$, then solution of given equation is equal to*

$$x(t) = C_q(t)x_0 + K_q(t)x_1 + \frac{1}{\Gamma\alpha}\int_0^t (t - s)^{\alpha-1}f(s, x(s))ds. \tag{65}$$

*Let the function $f$ in above equation satisfy following Krasnoselskii-Krein type conditions:*

$(H_1)$ $d(f(t, x), f(t, y)) \leqslant \Gamma(q)\frac{\alpha(k-1)+1}{t^{\alpha}}d(x, y), t \neq 0$, where $k > 1$.

$(H_2)$ $d(f(t, x), f(t, y)) \leqslant \zeta d(x, y)^{\beta}$, where $\zeta$ is constant, $0 < \beta < 1$, and $k(1 - \beta) < 1$, for $(t, x), (t, y) \in \mathbb{R}$.

*Then approximations are given by*

$$x_{n+1}(t) = C_q(t)x_0 + K_q(t)x_1 + \frac{1}{\Gamma\alpha}\int_0^t (t - s)^{\alpha-1}f(s, x_n(s))ds, \tag{66}$$

*converges uniformly to a unique solution $x(t)$ of given equations on $\{0, \mu\}$ where*

$$\mu = \min\left\{c, \left(\frac{e\Gamma(1 + \alpha)}{J}\right)^{\frac{1}{\alpha}}\right\},$$

*$J$ is bound for $f$ on $\mathbb{R}$.*

## 6. Conclusions

The existence and uniqueness of the class of high-order fuzzy Krasnoselskii-Krein conditions are investigated in this paper. This is a fruitful field with a wide range of research projects that can lead to various applications and theories. In future projects, we hope to learn more about fuzzy fractional evolution problems. Using the Caputo derivative, we can discover uniqueness and existence with uncertainty. Future work could include expanding on the concept proposed in this mission, including observability, and generalizing other activities. This is an interesting area with a lot of study going on that could lead to a lot of different applications and theories. This is a path to which we want to invest considerable resources.

**Author Contributions:** R.S., A.U.K.N., M.B.J. and N.H.A. contributed equally to the writing of this paper. All authors have read and agreed to the published version of the manuscript.

**Funding:** This work is not funded by government or any private agency.

**Data Availability Statement:** Data is original and references are given where required.

**Acknowledgments:** The authors would like to acknowledge The University of Lahore, for the provision of the research plate form to complete this research work.

**Conflicts of Interest:** The authors declare that they have no known competing financial interest or personal relationships that could have appeared to influence the work reported in this paper.

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
