# Peer review of "Existence and Uniqueness of Mild Solution Where α ∈ (1,2) for Fuzzy Fractional Evolution Equations with Uncertainty"

_fractalfract, doi:10.3390/fractalfract6020065_

Round 1

Reviewer 1 Report

The authors proved the existence and uniqueness of the mild solution for fuzzy fractional evolution equations with uncertainty. The paper is in general well written and the results are not available in the literature. Thus I may suggest the acceptance. Some minor notes:

  1. What is the meaning of the notation H in the fractional derivative?
  2. Some typos: In the abstract, C_\rho denote -> C_\rho denotes; Page 3, the symbol ' denotes -> the symbol $\prime$ denotes.
  3. Is it possible to extend the developed methods to variable-order fractional  problems? In particular, in arXiv: 2110.00752 and arXiv:2110.04707, the variable-order fractional Cauchy problem is transformed into the equivalent integral equation, which extends the existing results in constant-order problems.  Thus it is potential to extend the developed methods in this paper to this variable-order problem by using this equivalent integral equation. Some more references on the variable-order problem could be found in  https://doi.org/10.1023/A:1016586905654 https://doi.org/10.1080/10652469308819027  https://doi.org/10.1137/19M1245621  The authors could add a remark on this extension. 

Author Response

REVIEWER’s REPORT

On the paper

The authors proved the existence and uniqueness of the mild solution for fuzzy fractional evolution equations with uncertainty. The paper is in general well written and the results are not available in the literature. Thus I may suggest acceptance. Some minor notes:

  1. What is the meaning of the notation H in the Fractional derivative?

Ans: H is the Fractional derivative between (1,2).

  1. Some typos: In the abstract, C_\rho denote C_\rho denotes; Page 3, the symbol ‘ denotes the symbol $\prime$ denotes.

Ans: The change has been done.

  1. Is it possible to extend the developed methods to the variable-order fractional problem? In particular, in arXiv:2110.00752 and Xiv:2110.04707, the variable-order fractional Cauchy problem is transformed into the equivalent integral equation, which extends the existing results in constant-order problems. Thus it is potential to extend the developed methods in this paper to this variable-order problem by using this equivalent integral equation. Some more references on the variable-order problem could be found in https://doi.org/10.1023/A:1016586905654

https://doi.org/10.1080/10652469308819027

https://doi.org/10.1137/19M1245621

The authors could add a remark on this extension.

         Ans: Yes, it is possible to extend the developed methods to the variable-order fractional problem.

Reviewer 2 Report

Review report:
Submitted to: Fractal Fract (ISSN 2504-3110)
Manuscript ID fractalfract-1552439
Title of the paper
Local and global existence and uniqueness of   2 (1; 2) for a class of
fuzzy fractional functional evolution equation
Ramsha Shafqat
Reviewer comments
In this article, the author investigate fuzzy fractional functional evolution equa-
tions with fuzzy population models and distributed delays using fuzzy fractional
functional evolution equations. To explain these results, some theorems are given.
Finally, a few examples of fuzzy fractional functional evolution equations are shown
The results are new and very interesting for the large community of researchers
working in the  eld of mathematics and fractional calculus. A MINOR revision is
required to make the manuscript worth publishing.
 Revised the title if possible.
 Authors list in the paper is missing.
 Equation numbers are missing for some Equations.
 Give more details in the introduction part about the method.
 Please check carefully all the equation at the end (.) and (,) are missing.
 Page 4, put (,) at the end of Eq. 1.
 Page 8, in De nition 3.8 put (,) at the end of Eq.
 Page 10, put (.) at the end of Eq. 5.
 The author exempli ed the results which is good for the article.
 Overall presentation of the paper needs to improve.
 Remove unnecessary references.
 The authors are requested to check some latest work related to the existence
and uniqueness of solutions as below
https://doi.org/10.1155/2021/3297562, https://doi.org/10.1186/s13662-015-
0651-z

Author Response

Reviewer comments
In this article, the author investigates fuzzy fractional functional evolution equations with fuzzy population models and distributed delays using fuzzy fractional
functional evolution equations. To explain these results, some theorems are given.
Finally, a few examples of fuzzy fractional functional evolution equations are shown. The results are new and very interesting for the large community of researchers
working in the field of mathematics and fractional calculus. A MINOR revision is
required to make the manuscript worth publishing.
Revised the title if possible.

Ans: The change has been done.
Author’s list in the paper is missing.

Ans: The change has been done.
Equation numbers are missing for some Equations.

Ans: The change has been done.
Give more details in the introduction part about the method.

Ans: The change has been done.
Please check carefully all the equations at the end (.) and (,) are missing.

Ans: The change has been done.
Page 4, put (,) at the end of Eq. 1.

Ans: The change has been done.
Page 8, in Definition 3.8 put (,) at the end of Eq.

Ans: The change has been done.
Page 10, put (.) at the end of Eq. 5.

Ans: The change has been done.
The author exemplified the results which are good for the article.

Ans: The change has been done.
Overall presentation of the paper needs to improve.

Ans: The change has been done.
Remove unnecessary references.

Ans: The change has been done.
The authors are requested to check some latest work related to the existence
and uniqueness of solutions as below
https://doi.org/10.1155/2021/3297562, https://doi.org/10.1186/s13662-015-
0651-z

Ans: The change has been done.

Reviewer 3 Report

REVIEWER’s REPORT

On the paper

Local and global existence and uniqueness of α (1,2) for a class of fuzzy fractional functional evolution equation

The author should consider the following points while revising this paper:

  1. This paper's motivation isn't clear enough. Please explain what is new, the impact of your work, and how it will make a difference.
  2. The abstract needs to be rewritten.
  3. This study's contributions should be explained in greater detail. I propose that further in-depth discussions of past efforts in the literature be performed to emphasize the contributions.
  4. The introduction part should be included the significance and novelty of the paper.
  5. Many typos exist that should be checked and corrected throughout the paper.
  6. Grammatical errors exist that should be checked and corrected throughout the paper.
  7. In order to show the efficiency of the method, the comparison result with the existing method can be provided.
  8. I am not able to understand the application part. How these examples are connected with the discussed theory part. Need more explanations.
  9. The reference section should be improved by adding the recent papers published related to this work.

                             Considering the above points, I recommend the paper for a MAJOR revision to the journal Fractal and Fractional.

Author Response

REVIEWER’s REPORT

The author should consider the following points while revising this paper:

  1. This paper's motivation isn't clear enough. Please explain what is new, the impact of your work, and how it will make a difference.

Ans: This paper explains the existence and uniqueness of a mild solution for a fuzzy fractional evolution equation with uncertainty.

  1. The abstract needs to be rewritten.

Ans: The change has been done.

  1. This study's contributions should be explained in greater detail. I propose that further in-depth discussions of past efforts in the literature be performed to emphasize the contributions.

Ans: The change has been done.

  1. The introduction part should be included the significance and novelty of the paper.

Ans: The change has been done.

  1. Many typos exist that should be checked and corrected throughout the paper.

Ans: The change has been done.

  1. Grammatical errors exist that should be checked and corrected throughout the paper.

Ans: The change has been done.

  1. In order to show the efficiency of the method, the comparison result with the existing method can be provided.

Ans: The change has been done.

  1. I am not able to understand the application part. How these examples are connected with the discussed theory part. Need more explanations.

Ans: The change has been done.

  1. The reference section should be improved by adding the recent papers published related to this work.

Ans: The change has been done.

Round 2

Reviewer 3 Report

  1. The abstract should be rewritten. Check the last line. Still, I am not satisfied with the abstract part.
  2. The authors are advised to use standard notations.
  3. In page 2: lines 11-14, not clear.
  4. The introduction part needs to rewrite. Not satisfied with the introduction part too.
  5. In page 2: line -8: it is [] or ()?
  6. The application part is still not clear. How the authors connect this application part with the theory part.
  7. The conclusion needs to be rewritten.

Based on the above points, I suggest MINOR revision this time.

Author Response

REVIEWER’s REPORT

                                              Comments and Suggestions for Authors

  1. The abstract should be rewritten. Check the last line. Still, I am not satisfied with the abstract part.

Ans: The change has been done.

  1. The authors are advised to use standard notations.

Ans: The change has been done.

  1. In page 2: lines 11-14, not clear.

Ans: The change has been done.

  1. The introduction part needs to rewrite. Not satisfied with the introduction part too.

      Ans: The change has been done.

  1. In page 2: line -8: it is [] or ()?

Ans: It is [].

  1. The application part is still not clear. How the authors connect this application part with the theory part.

Ans: The application part has been changed.

  1. The conclusion needs to be rewritten.

Ans: The change has been done.
